# RSS-Based Target Localization in Underwater Acoustic Sensor Networks via Convex Relaxation

**DOI:** 10.3390/s19102323

**Published:** 2019-05-20

**Authors:** Shengming Chang, Youming Li, Yucheng He, Yongqing Wu

**Affiliations:** 1Faculty of Electrical Engineering and Computer Science, Ningbo University, Ningbo 315211, China; 2School of Electronic and Information Engineering, Ningbo University of Technology, Ningbo 315211, China; 3Xiamen Key Laboratory of Mobile Multimedia Communications, Huaqiao University, 668 Jimei Avenue, Xiamen 361021, China; yucheng.he@hqu.edu.cn; 4Institute of Acoustics, Chinese Academy of Science, Beijing 100190, China; wyq@mail.ioa.ac.cn

**Keywords:** underwater acoustic wireless sensor networks (UWSNs), target localization, received signal strength (RSS), Cramer–Rao lower bounds (CRLBs)

## Abstract

The received signal strength (RSS) based target localization problem in underwater acoustic wireless sensor networks (UWSNs) is considered. Two cases with respect to target transmit power are considered. For the first case, under the assumption that the reference of the target transmit power is known, we derive a novel weighted least squares (WLS) estimator by using an approximation to the RSS expressions, and then transform the originally non-convex problem into a mixed semi-definite programming/second-order cone programming (SD/SOCP) problem for reaching an efficient solution. For the second case, there is no knowledge on the target transmit power, and we treat the reference power as an additional unknown parameter. In this case, we formulate a WLS estimator by using a further approximation, and present an iterative ML and mixed SD/SOCP algorithm for solving the derived WLS problem. For both cases, we also derive the closed form expressions of the Cramer–Rao Lower Bounds (CRLBs) on root mean square error (RMSE). Computer simulation results show the superior performance of the proposed methods over the existing ones in the underwater acoustic environment.

## 1. Introduction

### 1.1. Background

The target localization techniques based on an underwater acoustic wireless sensor networks (UWSNs) have recently attracted much attention due to their wide applications in many areas, including data collection, pollution monitoring, offshore exploration, disaster prevention, target tracking, and assisted navigation [1]. In such UWSNs, the sensor nodes are usually classified into anchor nodes and target nodes, where the locations of anchor nodes are known, and the locations of target nodes are unknown and need to be determined [2]. The target localization in UWSNs is often classified as four types based on the measurements: time-of-arrival (ToA), time-difference-of-arrival (TDoA), angle-of-arrival (AoA), and received signal strength (RSS). However, the measurements based on the first three types of target localization methods require complicated timing, synchronization, and line of sight, thus are often difficult to be obtained in underwater acoustic environments [3,4].

Currently, most works are focused on two typical RSS-based target localization methods: the maximum likelihood (ML) [5] and least squares (LS) [6]. While the ML estimator can achieve high accuracy and approximate the Cramer–Rao lower bound (CRLB), its performance depends highly on the initial point and a poor initialization may lead ML estimator to find a poor solution. To overcome this problem, the LS estimator with explicit solution is proposed at lower complexity. The weighted least squares (WLS) approach is more appropriate for estimating the source location and its solution is closer to the ML estimator. Therefore, the RSS-based LS and WLS estimator have attracted more attention [6,7].

### 1.2. Related Work

Recently, convex optimization techniques have been widely applied to the target localization. The basic idea is to transform the objective function into a new convex problem via semi-definite programming (SDP) or second-order cone programming (SOCP) relaxations, and then efficiently find solution which is approaching the globally optimal solution. In [8], to circumvent the non-convexity of the ML estimation, the ML estimation problem is reformulated by eliminating the logarithmic terms and relaxed the problem as a SDP optimization problem. In [9], the non-convex ML estimation problem is converted into an alternative SOCP optimization problem. Furthermore, in [10], a hybrid method by combining the RSS and AoA measurements is proposed to formulate a non-convex estimator under the LS criterion, and then transform it into a SOCP optimization problem [11]. However, all the approaches above are focused on the target localization problem in terrestrial wireless sensor networks. In [12], a convex optimization method for target localization in UWSNs is investigated, but still under a terrestrial acoustic wave propagation model. In [13], the ML problem for the RSS-based underwater target localization are discussed and a class of SDP methods are derived by using the $\ell_{1}$ norm instead of $\ell_{2}$ norm. In [14], RSS-based underwater target localization methods are proposed in both known and unknown transmit power cases, and two fast implementation algorithms are proposed by transforming the non-convex problems into generalized trust region subproblem frameworks. All of the above methods are simple to solve the underwater RSS localization problem. However, their results show that there still exists great room for the estimation accuracy improvement.

### 1.3. Contributions

In this paper, we propose a new approach to the RSS-based underwater acoustic localization problem based on the convex relaxation technique in UWSNs. Unlike the SDP method in [13] and WLS methods in [14], we reformulate the original underwater acoustic path loss measurement model as the pseudolinear equation, derive a minimum optimization problem, and then directly transform this optimization problem into a mixed SD/SOCP localization problem for estimating an accurate location of target. Although the WLS methods in [14] can simply solve the underwater RSS localization problem with lower complexity, its accuracy cannot be perfect. This paper is to improve the underwater acoustic RSS localization accuracy. Two cases are considered with respect to the target transmit power. For the first case, where the reference power is assumed to be known as a measure of the target transmit power, we derive a novel WLS estimator by using an approximation to the RSS expressions, and then transform the originally non-convex problem into a mixed SD/SOCP problem for reaching an efficient solution. For the second case, there is no knowledge on the target transmit power, and we treat the reference power as an additional parameter to estimate. In this case, we formulate a WLS estimator by using a further approximation, and present an iterative ML and mixed SD/SOCP algorithm for solving the WLS problem. For both cases, we also derive the Cramer–Rao Lower Rounds (CRLBs) on root mean square error (RMSE) to evaluate the performance of the two proposed WLS estimators by using computer simulations.

The main contributions of our work are summarized as follows:(1)Based on the convex relaxation, we propose a new approach to the RSS-based underwater acoustic localization problem in UWSNs.(2)The correlation between the target node location and transmit power can be removed by using the separation constant technique which will be helpful for solving the proposed localization method.(3)In the unknown target transmit power case, an auxiliary constant is used to establish the semi-definite constraints., and then an iterative ML and mixed SD/SOCP algorithm is proposed to estimate both the target location and the transmit power.(4)The closed form CRLBs of the proposed scheme are derived.

The following notations are adopted throughout the paper. Bold face lower case letters and bold face upper case letters denote vectors and matrices, respectively. $R^{n}$ denotes the set of *n*-dimensional real column vectors. $l_{i}$ denotes the *i*th entry of the vector l. In addition, ∥·∥ denotes the $\ell_{2}$-norm.

The remainder of the paper is organized as follows. In Section 2, we first give the ray trajectory in an underwater medium, and then the RSS models for underwater acoustic node localization are discussed. In Section 3, we present the proposed localization method. In Section 4, the Cramer–Rao Lower Rounds are derived for the proposed method. Computer simulation results are presented in Section 5. Finally, the main conclusions are drawn in Section 6.

## 2. System Model

In this section, we first given the arc length of tracing a ray between a target node (denoted as T) and the anchor nodes (denoted as A) in an underwater environment [15,16]. Then, the RSS-based localization problem in underwater acoustics environment is formulated.

### 2.1. Ray Trajectory

We consider a target localization problem in a 3D underwater acoustic environment. The main considered in this paper are the stratification effect of water medium. Aiming at the stratification effect of underwater environment, in this paper, we propose a new approach to the RSS-based underwater acoustic localization problem based on the convex relaxation technique in UWSNs. In this approach, the underwater sound speed profile (SSP) is assumed only linearly depth dependent and can be approximated as [15,16,17](1)v(z)=b+az,
where *z* denotes the depth, *b* indicates the sound speed at the water surface, and *a* is the steepness of SSP depending on the environment of the stratification effect of water medium. Let $\left[x_{A},y_{A},z_{A}\right]$ and $\left[x_{T},y_{T},z_{T}\right]$ denote $\left[r_{A},z_{A}\right]$ and $\left[r_{T},z_{T}\right]$, respectively, in 3D space. In Figure 1, we can seen that the UWSNs is 3D, while the ray equations are established on a 2D plane. This is because of the cylindrical symmetry around the *z*-axis when the *z*-axis crosses anchor node A. In this case, we can transfer the target localization problem to the 2D plane, which includes both nodes and *z*-axis. When the *z*-axis does not cross anchor node A, we still consider it to be a general 3D space.

Equation (Equation 1) reveals that the SSP is only a linear function of the depth, but the SSP and horizontal distance *r* are nonlinear relations. Ray tracing method is guided by Snell’s law [17]:(2)$$\frac{\cos\theta }{v(z)}=\frac{\cos\theta_{A}}{v(z_{A})}=\frac{\cos\theta_{T}}{v(z_{T})}=c,$$ where $\theta_{A},\theta_{T}$ are the ray angles at anchor node and the target node locations, respectively; $z_{A},z_{T}$ are the depth of the anchor node and target node, respectively; and *c* is constant along a ray traveling between the two nodes. Then, the arc length *l* of acoustic propagation path between the anchor node and target node has the following form [15,16](3)$$l=-\left(b+az_{T}\right)\frac{\theta_{T}-\theta_{A}}{acos\theta_{T}},$$

In the next section, we suppose that the speed in each layer is assumed to be a constant. In this case, the arc length *l* is assumed to be approximated as a straight line, which is used in the following discussion.

### 2.2. Underwater Acoustic RSS Model

The main characteristics of underwater acoustic channel are frequency-dependent attenuation, time-varying multipath propagation, and low speed of sound. Up to now, the underwater acoustic channel is considered the most difficult communication media. Underwater acoustic transmission loss experienced by a narrow band acoustic signal centered at frequency *f* (kHz) traveling over distance *l* is given by Urick propagation model, and then it can be transformed into RSS model. In the following, we describe the signal model considered in this paper.

Consider an UWSNs with *N* anchor nodes and one target node. For a *k*-dimensional (k=2 or 3) localization scenario, we suppose that the target node is located at an unknown position $x\in R^{k}$, and the locations of anchor nodes are known as $s_{i}\in R^{k}$, where i∈N≜{1,2,…,N}. Under a centralized processing mode, the anchor nodes convey their RSS measurements to the central processor for estimating x.

By the underwater acoustic propagation log-normal shadowing model, the RSS in dBm at each anchor node *i* is given by [13,14,18](4)$$P_{i}=P_{0}-10\gamma \log_{10}\frac{l_{i}}{l_{0}}-\alpha_{f}\left(l_{i}-l_{0}\right)+n_{i},$$
where $P_{0}$ is the reference power, $d_{0}$ is the reference distance, and $l_{i}$ is the ray length between the target node and *i*th anchor node, which can be calculated by Equation (Equation 3). For the same layer of acoustic speed, the arc length between the target node and *i*th anchor node is approximated to $l_{i}=∥x-s_{i}∥$, γ is the path-loss exponent between 2 and 4 depending upon the propagation environment, $\alpha_{f}$ is the absorption coefficient, and $n_{i}$ is the log-normal shadowing effects [19], which results from degradation of the acoustic intensity caused by multipath propagation, refraction, diffraction, and scattering of sound. Note that $\alpha_{f}$ can be obtained in dB/km as a function of signal frequency *f* by Thorp’s formula [20,21]    (5)$$a_{f}=\frac{0.11f^{2}}{1+f^{2}}+\frac{44f^{2}}{4100+f^{2}}+2.75\times 10^{-4}f^{2}+0.003.$$

Let $P={\left(P_{1},P_{2},\ldots ,P_{N}\right)}^{T}$ and $n={\left(n_{1},n_{2},\ldots ,n_{N}\right)}^{T}$ be the collections of the RSS measurements and the associated log-normal shadowing effects, respectively. In most studies, the log-normal shadowing is generally modeled as independent and identical distribution (i.i.d.)zero-mean Gaussian random variable. Unfortunately, we do not have such a guarantee about $n_{i}$ in underwater acoustic environment. In this paper, we suppose that the N×N covariance matrix Q of n is symmetric with elements given by(6)$${\left[Q\right]}_{ij}=\left\{\begin{array}{cc} \sigma^{2}, & i=j,\\ \rho \sigma^{2}, & i\neq j, \end{array}\right.$$
where 0≤ρ<1 is the common correlation coefficient between any two different entries.

Given the target location x, the conditional probability density function (PDF) of P is given by [18,22](7)$$\begin{array}{c} f\left(P|x\right)=\frac{1}{{\left(2\pi \right)}^{\frac{N}{2}}{\left|Q\right|}^{\frac{1}{2}}}e^{-\frac{1}{2}{\left[P-\bar{P}\left(x\right)\right]}^{T}Q^{-1}\left[P-\bar{P}\left(x\right)\right]}, \end{array}$$
where $\bar{P}\left(x\right)={\left({\bar{P}}_{1}\left(x\right),{\bar{P}}_{2}\left(x\right),\ldots ,{\bar{P}}_{N}\left(x\right)\right)}^{T}$, and for all i∈N,(8)$$\begin{array}{cc} {\bar{P}}_{i}\left(x\right)=\; & P_{0}-10\gamma \log_{10}\frac{∥x-s_{i}∥}{d_{0}}-\alpha_{f}\left(∥x-s_{i}∥-d_{0}\right). \end{array}$$

Then, the ML estimator of x can be formulated as [22](9)$$\begin{array}{c} \hat{x}=\underset{x}{\mathrm{error}}\;{\left[P-\bar{P}\left(x\right)\right]}^{T}Q^{-1}\left[P-\bar{P}\left(x\right)\right]. \end{array}$$

Clearly, the ML estimation, whether in Equation (Equation 9), yields a non-convex problem. In the next section, we propose a mixed SD/SOCP method for efficiently solving the localization problem by using appropriate approximations for the RSS expressions.

## 3. The Proposed Mixed SD/SOCP Methods

### 3.1. The WLS Estimation by Approximation

Suppose that the noises $n_{i}$ are quite small when compared with the RSS measurements $P_{i}$. Let $l_{0}=1$ m without loss of generality. From Equation (Equation 4), we have the following approximation expression(10)$$\begin{array}{c} ∥x-s_{i}∥\approx 10^{\frac{P_{0}-P_{i}+\alpha_{f}}{10\gamma }}10^{-\frac{\alpha_{f}∥x-s_{i}∥}{10\gamma }}. \end{array}$$

Let $u=-\frac{\alpha_{f}∥x-s_{i}∥}{10\gamma }$ be the exponent of the second factor term on the right hand side of Equation (Equation 10). It is shown in [13] that 0<|u|≪1 since $\alpha_{f}$ is relatively minor in UWSNs, as shown in Equation (Equation 5). Therefore, it is reasonable to approximate the factor term $10^{u}$ to be its first-order Taylor expansion at the point u=0, i.e.,(11)$$\begin{array}{c} 10^{-\frac{\alpha_{f}∥x-s_{i}∥}{10\gamma }}\approx 1-\frac{\alpha_{f}\ln10}{10\gamma }∥x-s_{i}∥, \end{array}$$
where the higher-order terms are omitted.

Substituting Equation (Equation 11) into Equation (Equation 10) yields a pseudolinear approximation(12)$$\begin{array}{c} \lambda_{i}∥x-s_{i}∥\approx \eta , \end{array}$$
where $\eta =10^{\frac{P_{0}}{10\gamma }}$, $\beta_{i}=10^{\frac{P_{i}-\alpha_{f}}{10\gamma }}$, and $\lambda_{i}=\beta_{i}+\frac{\alpha_{f}\ln10}{10\gamma }\eta$.

Based on Equation (Equation 12), the WLS estimation for the target location x can be formulated as(13)$$\begin{array}{cc} \min_{x}\; & \sum_{i=1}^{N}(\lambda_{i}∥x-s_{i}{∥-\eta )}^{T}Q^{-1}(\lambda_{i}∥x-s_{i}∥-\eta ) \end{array}$$
where ${\left\{\eta ,\lambda_{i}\right\}}_{i=1}^{N}$ are dependent upon both the reference power $P_{0}$ and the RSS parameters ${\left\{\gamma ,Q,\alpha_{f},s_{i},P_{i}\right\}}_{i=1}^{N}$. Clearly, the proposed WLS estimation is non-convex.

### 3.2. Case of Known Target Transmit Power

In this case, we suppose that the transmit power of the target node is known in terms of the reference power $P_{0}$. It follows that the parameters ${\left\{\eta ,\lambda_{i}\right\}}_{i=1}^{N}$ in Equation (Equation 13) can be determined.

For the sake of presentation, by introducing auxiliary variables $l,l_{i}$, the WLS estimation (Equation (Equation 13)) can be equivalently written as(14a)$$\begin{array}{cc} \min_{x}\; & {\left(Bl-E\right)}^{T}Q^{-1}\left(Bl-E\right)\\ s.t.\;\;\; & l={\left[l_{1},\ldots ,l_{N}\right]}^{T}, \end{array}$$
(14b)$$\begin{array}{c} l_{i}=∥x-s_{i}∥,i\in N, \end{array}$$
where $B=diag\left\{\lambda_{1},\cdots ,\lambda_{N}\right\},E={\left[\eta ,\cdots ,\eta \right]}^{T}$, Q is the covariance matrix of log-normal shadowing effect.

Then, by further introducing auxiliary variables {L,r}, and applying the SDP technique, the WLS estimation (Equation (14)) can be rewritten in the constricted form as(15a)$$\begin{array}{cc} \min_{x}\; & tr\left\{C\left[\begin{array}{cc} L & l\\ l^{T} & 1 \end{array}\right]\right\}\\ s.t.\;\;\; & l_{i}=∥x-s_{i}∥, \end{array}$$
(15b)$$\begin{array}{c} l_{i}^{2}=L_{i,i}, \end{array}$$
(15c)$$\begin{array}{c} L_{i,i}=r-2s_{i}^{T}x+{∥s_{i}∥}^{2}, \end{array}$$
(15d)$$\begin{array}{c} \left[\begin{array}{cc} L & l\\ l^{T} & 1 \end{array}\right]⪰0,rank\left\{D\right\}=1, \end{array}$$
(15e)$$\begin{array}{c} {∥x∥}^{2}=r, \end{array}$$
$$\begin{array}{c} \mathrm{for}\;\mathrm{all}\;i\in N, \end{array}$$
where(16)$$C=\left[\begin{array}{cc} B^{T}Q^{-1}B & -B^{T}Q^{-1}E\\ -E^{T}Q^{-1}B & E^{T}Q^{-1}E \end{array}\right].$$

Furthermore, according to the Cauchy–Schwarz inequality, we have(17)$$\begin{array}{cc} L_{i,j} & \geq |r-{\left(s_{i}+s_{j}\right)}^{T}x+s_{i}^{T}s_{j}|,\\ & \mathrm{for}\;\mathrm{all}\;i,j\in N,i>j. \end{array}$$

Adding Equation (17) into Equation (15) as a constraint, dropping rank-1 constraint, and applying the SOCP technique, the Equation (15) can be transformed to be    (18a)$$\begin{array}{cc} \min_{x}\; & tr\left\{C\left[\begin{array}{cc} L & l\\ l^{T} & 1 \end{array}\right]\right\}\\ s.t.\;\;\; & ∥x-s_{i}∥\leq l_{i}, \end{array}$$
(18b)$$\begin{array}{c} l_{i}^{2}\leq L_{i,i}, \end{array}$$
(18c)$$\begin{array}{c} L_{i,i}=r-2s_{i}^{T}x+{∥s_{i}∥}^{2}, \end{array}$$
(18d)$$\begin{array}{c} L_{i,j}\geq \left|r-{\left(s_{i}+s_{j}\right)}^{T}x+s_{i}^{T}s_{j}\right|, \end{array}$$
(18e)$$\begin{array}{c} \left[\begin{array}{cc} L & l\\ l^{T} & 1 \end{array}\right]⪰0, \end{array}$$
(18f)$$\begin{array}{c} {∥x∥}^{2}\leq r, \end{array}$$
$$\begin{array}{c} \mathrm{for}\;\mathrm{all}\;i,j\in N,i>j, \end{array}$$
where the constraint in Equation (18a) is a second-order cone (SOC), the constraint in Equation (18e) is a semi-definite cone (SDC), the constraints in Equations (18c) and (18d) are affine and thus SOCs, and the constraints in Equations (15c) and (18f) need to be reformulated into SOC forms.

It is observed that the constraints in Equations (18b) and (18f) have a unified form of $\frac{{∥y∥}^{2}}{a}=b$, which can be expressed in the second-order cone form of $∥[2y;a-b]∥\leq a+b$, where [y;e] denotes the concatenation of column vector y and scalar *e* [23]. Therefore, Equation (15) can be casted into a mixed SD/SOCP problem with the optimization variables {x,r,L,l} as follows(19a)$$\begin{array}{cc} \min_{x}\; & tr\left\{C\left[\begin{array}{cc} L & l\\ l^{T} & 1 \end{array}\right]\right\}\\ s.t.\;\;\; & ∥x-s_{i}∥\leq l_{i}, \end{array}$$
(19b)$$\begin{array}{c} ∥\left[2l_{i};L_{i,i}-1\right]∥\leq L_{i,i}+1, \end{array}$$
(19c)$$\begin{array}{c} L_{i,i}=r-2s_{i}^{T}x+{∥s_{i}∥}^{2}, \end{array}$$
(19d)$$\begin{array}{c} L_{i,j}\geq \left|r-{\left(s_{i}+s_{j}\right)}^{T}x+s_{i}^{T}s_{j}\right|, \end{array}$$
(19e)$$\begin{array}{c} \left[\begin{array}{cc} L & l\\ l^{T} & 1 \end{array}\right]⪰0, \end{array}$$
(19f)$$\begin{array}{c} ∥[2x;r-1]∥\leq r+1, \end{array}$$
$$\begin{array}{c} \mathrm{for}\;\mathrm{all}\;i,j\in N,i>j. \end{array}$$

In the sequel, the mixed SD/SOCP problem in Equation (19) is referred to as “SD/SOCP-K”, which can be solved efficiently by using the standard interior method [24], for example, the MATLAB CVX package [25]. Since there exist $K_{1}=5$ optimization variables in Equation (19), the *worst-case* complexity has order of $O({\left(N+K_{1}\right)}^{3.5})$, which is less than that for solving a SDP problem [13].

### 3.3. Case of Unknown Target Transmit Power

In practice, it is impractical to acquire the knowledge on the transmit power of the target node in UWSNs. It is thus worth investigating the localization problem with unknown $P_{0}$. To cope with the difficulty caused by the unknown parameters, we explore a further approximation in addition to Equation (Equation 12).

By the definitions of $\beta_{i}$ and η, using the separation constant technique, the approximation in Equation (Equation 12) can be reorganized as(20)$$\begin{array}{c} \frac{10\gamma \beta_{i}}{\alpha_{f}\ln10}\left(\frac{1}{1-\frac{\alpha_{f}\ln10}{10\gamma }∥x-s_{i}∥}-1\right)\approx \eta . \end{array}$$

Let $v=\frac{\alpha_{f}\ln10}{10\gamma }∥x-s_{i}∥$. Following the explanations beneath Equation (Equation 10), it also holds that 0<|v|≪1. It is thus reasonable to approximate the factor in parentheses on the left hand side by using the first-order Taylor expansion about v=0 as follows(21)$$\begin{array}{c} \frac{1}{1-\frac{\alpha_{f}\ln10}{10\gamma }∥x-s_{i}∥}\approx 1+\frac{\alpha_{f}\ln10}{10\gamma }∥x-s_{i}∥. \end{array}$$

Substituting Equation (21) into Equation (20) yields a simplified approximation(22)$$\begin{array}{c} \beta_{i}∥x-s_{i}∥\approx \eta . \end{array}$$

Hence, instead of solving Equation (Equation 13), we obtain a WLS formulation for estimating both x and $P_{0}$ in the form(23)$$\begin{array}{c} \min_{x,\eta }\;\sum_{i=1}^{N}(\beta_{i}∥x-s_{i}{∥-\eta )}^{T}Q^{-1}(\beta_{i}∥x-s_{i}∥-\eta ). \end{array}$$
where again $\eta =10^{\frac{P_{0}}{10\gamma }}$, and $\beta_{i}=10^{\frac{P_{i}-\alpha_{f}}{10\gamma }}$.

Clearly, the WLS problem in Equation (23) remains non-convex. To cope with the objective function, we introduce an auxiliary vector $g={\left(g_{1},g_{2},\ldots ,g_{N},\eta +c_{0}\right)}^{T}$, where $g_{i}=∥x-s_{i}∥,g_{N+1}=\eta +c_{0}$, $c_{0}$ is an arbitrary constant, such that the objective function is written as ${\left(\tilde{B}g-V\right)}^{T}Q^{-1}\left(\tilde{B}g-V\right)$. Then, we transform the WLS problem in Equation (23) into the following constricted form(24a)$$\begin{array}{cc} \min_{x,\eta }\; & {\left(\tilde{B}g-V\right)}^{T}Q^{-1}\left(\tilde{B}g-V\right)\\ s.t.\;\;\; & g=[g_{1},g_{2},\ldots ,g_{N},g_{N+1}{]}^{T}, \end{array}$$
(24b)$$\begin{array}{c} g_{i}=∥x-s_{i}∥, \end{array}$$
(24c)$$\begin{array}{c} g_{N+1}=\eta +c_{0}, \end{array}$$
$$\begin{array}{c} \mathrm{for}\;\mathrm{all}\;i\in N, \end{array}$$
where $\tilde{B}=\left[diag\left(\beta_{i}\right),-1_{N}\right]$, $V=-c_{0}1_{N}$, $1_{N}$ is N×1 column vector. Q is covariance matrix of log-normal shadowing effect.

By further introducing auxiliary variables {G,r}, and applying the SDP technique, we transform the WLS problem in Equation (24) into the following constricted form (25a)$$\begin{array}{cc} \min_{x,\eta }\; & tr\left\{\tilde{C}\left[\begin{array}{cc} G & g\\ g^{T} & 1 \end{array}\right]\right\}\\ s.t.\;\;\; & g_{i}=∥x-s_{i}∥, \end{array}$$
(25b)$$\begin{array}{c} G_{i,i}=r-2s_{i}^{T}x+{∥s_{i}∥}^{2}, \end{array}$$
(25c)$$\begin{array}{c} g_{i}^{2}=G_{i,i}, \end{array}$$
(25d)$$\begin{array}{c} g_{N+1}=\eta +c_{0}, \end{array}$$
(25e)$$\begin{array}{c} g_{N+1}^{2}=G_{N+1,N+1}, \end{array}$$
(25f)$$\begin{array}{c} \left[\begin{array}{cc} G & g\\ g^{T} & 1 \end{array}\right]⪰0,rank\left\{G\right\}=1, \end{array}$$
(25g)$$\begin{array}{c} {∥x∥}^{2}=r, \end{array}$$
$$\begin{array}{c} \mathrm{for}\;\mathrm{all}\;i\in N, \end{array}$$
where(26)$$\tilde{C}=\left[\begin{array}{cc} {\tilde{B}}^{T}Q^{-1}\tilde{B} & -{\tilde{B}}^{T}Q^{-1}V\\ -V^{T}Q^{-1}\tilde{B} & V^{T}Q^{-1}V \end{array}\right].$$

Furthermore, according to the Cauchy–Schwarz inequality, we also have(27)$$\begin{array}{cc} G_{i,j} & \geq |r-{\left(s_{i}+s_{j}\right)}^{T}x+s_{i}^{T}s_{j}|,\\ & \mathrm{for}\;\mathrm{all}\;i,j\in N,i>j, \end{array}$$

Adding Equation (27) into Equation (25) as a constraint, and dropping rank-1 constraint, by the reformulations above, the non-convex optimization problem is casted into a mixed SD/SOCP with the optimization variables {x,η,r,G,g} as follows(28a)$$\begin{array}{cc} \min_{x,\eta }\; & tr\left\{\tilde{C}\left[\begin{array}{cc} G & g\\ g^{T} & 1 \end{array}\right]\right\}.\\ s.t.\;\;\; & ∥x-s_{i}∥\leq g_{i}, \end{array}$$
(28b)$$\begin{array}{c} G_{i,i}=r-2s_{i}^{T}x+{∥s_{i}∥}^{2}, \end{array}$$
(28c)$$\begin{array}{c} G_{i,j}\geq \left|r-{\left(s_{i}+s_{j}\right)}^{T}x+s_{i}^{T}s_{j}\right|, \end{array}$$
(28d)$$\begin{array}{c} ∥\left[2g_{i};G_{i,i}-1\right]∥\leq G_{i,i}+1, \end{array}$$
(28e)$$\begin{array}{c} ∥\left[2g_{N+1};G_{N+1,N+1}-1\right]∥\leq G_{N+1,N+1}+1, \end{array}$$
(28f)$$\begin{array}{c} \left[\begin{array}{cc} G & g\\ g^{T} & 1 \end{array}\right]⪰0, \end{array}$$
(28g)$$\begin{array}{c} ∥[2x;r-1]∥\leq r+1, \end{array}$$
$$\begin{array}{c} \mathrm{for}\;\mathrm{all}\;i,j\in N,i>j. \end{array}$$

The mixed SD/SOCP problem in Equation (28) is referred to as “SD/SOCP-U”, which can be solved by calling the MATLAB CVX package at a computational complexity of order $O({\left(N+K_{2}\right)}^{3.5})$, where $K_{2}=6$.

In Algorithm 1, we also propose a hybrid ML-SD/SOCP method by iteratively estimating $P_{0}$ and calling the standard CVX package for solving Equation (28), where the maximum iteration number $N_{\max}$ can be determined empirically with aid of numerical results on the convergence.

**Algorithm 1** hybrid ML-SD/SOCP
1:Initialize the iteration index n←0, and set the reference distance $l_{0}=1$ m2:find an initial estimate $\hat{x}$ in the feasible region of Equation (28)3:
**repeat**
4:
n←n+1
5:compute ${\hat{l}}_{i}\leftarrow ∥\hat{x}-s_{i}∥$ for all i∈N6:Use $\hat{x}$ to compute the ML estimate of $P_{0}$, ${\hat{P}}_{0}$ as$$\begin{array}{c} {\hat{P}}_{0}=\frac{1}{N}\sum_{i=1}^{N}\left[P_{i}+10\gamma \log_{10}{\hat{l}}_{i}+\alpha_{f}\left({\hat{l}}_{i}-l_{0}\right)\right] \end{array}$$7:compute $\hat{\eta }\leftarrow 10^{\frac{{\hat{P}}_{0}}{10\gamma }}$ and ${\hat{\lambda }}_{i}\leftarrow \beta_{i}+\frac{\alpha_{f}\ln10}{10\gamma }\hat{\eta }$ for all i∈N8:solve Equation (19), and obtain an updated $\hat{x}$9:**until** the estimates $\left\{\hat{x},\hat{\eta }\right\}$ converge $n=N_{\max}$10:**return**x and $P_{0}$ with $P_{0}=10\gamma \log_{10}\eta$


## 4. CRLB Analysis

For the target location x, the WLS estimator $\hat{x}$ in either Equation (19) or Equation (28) is an unbiased estimator, i.e., $E(\hat{x})=x$, as they are essentially based on Equation (Equation 9). Then, the covariance matrix of $\hat{x}$ is subject to the CRLB as $\mathrm{VAR}\left(\hat{x}\right)⪰F^{-1}$, where F is the Fisher information matrix (FIM).

To evaluate the accuracy performance of the unbiased estimation, the RMSE is defined as(29)$$\mathrm{RMSE}=\sqrt{\left(\frac{1}{M}\sum_{i=1}^{M}{∥{\hat{x}}_{i}-x_{i}∥}^{2}\right)},$$ where ${\hat{x}}_{i}$ is the estimate of the randomly generated target location $x_{i}$ in the *i*th simulation, and *M* is the number of independent simulation rounds.

Accordingly, we define the CRLB on RMSE by computing the root trace of $F^{-1}$. As proved in the Appendix A, the CRLBs on RMSE for both SD/SOCP methods are given, respectively, by(30)$$\begin{array}{c} {\mathrm{CRLB}}_{K}=\sqrt{\mathrm{trace}\left[{\left(G^{T}Q^{-1}G\right)}^{-1}\right]}, \end{array}$$
(31)$$\begin{array}{c} {\mathrm{CRLB}}_{U}=\sqrt{\mathrm{trace}\left[{\left(H^{T}Q^{-1}H\right)}^{-1}\right]}, \end{array}$$ where $G=\frac{\partial \bar{P}\left(x\right)}{\partial x}$, $H={\left(G,1_{N}\right)}_{N\times 3}$, and(32)$$\begin{array}{c} \frac{\partial {\bar{P}}_{i}\left(x\right)}{\partial x}=-\left(\frac{10\gamma }{\ln10∥x-s_{i}{∥}^{2}}+\frac{\alpha_{f}}{∥x-s_{i}∥}\right){\left(x-s_{i}\right)}^{T}, \end{array}$$ which is derived from Equation (Equation 8).

## 5. Simulation Results

In this section, computer simulation results are provided to evaluate the performance of the proposed mixed SD/SOCP methods, i.e., SD/SOCP-K and SD/SOCP-U. The RSS values were generated according to Equation (Equation 4) under the underwater acoustic propagation model, where $l_{0}=1$ m, $P_{0}=-40$ dBm, and γ=2, which is applicable to the scenario of underwater acoustic spherical spreading. We assumed that the depth of the target node could be obtained through pressure sensors, and, therefore, only the *x* and *y* coordinates of target location were estimated, which reduced the computational complexity of the proposed methods. The same hypothesis is also seen in [26]. We also assumed that only one layer of SSP was considered in the simulations. Then, the anchor and target nodes were randomly located within a square region of 100×100 m${}^{2}$, and the correlation coefficient was set to be ρ=0.8. A total of Mc=3000 Monte Carlo simulations were carried out. Here, target location x was a random variable of Monte Carlo estimation. The estimation of random variables x (target location) in each Monte Carlo simulation was independent of each other. For reader’s convenience, the simulation parameters are listed in Table 1.

Note that SD/SOCP-K was simulated by directly calling the CVX package [25,27], whereas SD/SOCP-U was simulated by using Algorithm 1. For comparison, simulation results on the SDP method in [13] and the WLS methods (including WLS-K and WLS-U) in [14] are provided. For convenience, the discussed estimators are listed in Table 2.

In Figure 2, we present one possible network configuration and the estimation accuracy of the target location in 2D (Figure 2a) and 3D (Figure 2b) for N=10, σ=4 dB, $\alpha_{f}=0.01$ dB/m. The figure shows that better estimation accuracy is achieved for target node.

Note that the acoustic signal attenuates dramatically in underwater acoustic channels, which depends upon both transmission range and frequency. In addition, underwater acoustic signal frequencies are available from tens of hertz to hundreds of kilohertz due to the severe frequency-dependent attenuation [28,29,30,31]. In simulations, we chose several frequency components for *f* within the range from 34 kHz to 454 kHz, and the associated absorption coefficient $\alpha_{f}$ varies from 0.01 dB/m to 0.1 dB/m by Equation (Equation 5). We considered four localization scenarios to evaluate the performance of the proposed methods in underwater acoustic environment.

Figure 3 shows the simulation results on the effects of the noise standard derivation σ on the RMSE for σ varying from 1 dB to 5 dB when $P_{0}$ is known (Figure 3a) and unknown (Figure 3b), where the number of anchor nodes is fixed at N=10, and the absorption coefficient is fixed at $\alpha_{f}=0.01$ dB/m. By the simulation setups, we had u=−0.05 and v=0.1151 for the approximations in Equations (Equation 11) and (21), respectively. In these figures, it is naturally observed that the RMSE increased with the noise level in all methods. It was also observed that both the proposed mixed SD/SOCP methods yielded smaller RMSE values than the SDP method and WLS method. Furthermore, Figure 3 shows the superior RMSE performance of the proposed methods SD/SOCP-K and SD/SOCP-U when σ is small, while the gap between the SD/SOCP methods and WLS methods become smaller when σ is large.

Figure 4 shows the simulation results on the effects of the number of anchor nodes *N* on the RMSE for 8≤N≤22, σ=4 dB, and $\alpha_{f}=0.01$ dB/m when $P_{0}$ is known (Figure 4a) and unknown (Figure 4b). In this case, it remained that u=−0.05 and v=0.1151. It was observed that the RMSE appears a decreasing function of *N* for both the proposed SD/SOCP methods and WLS methods. In contrast, the RMSE of the SDP method approximately increased with *N*, and it was considerably larger than that of the proposed SD/SOCP methods over the entire range of considered *N* values. The main reason for this lies in adopting more approximations and replacing the $\ell_{2}$ norm with the $\ell_{1}$ to solve the original localization problem of the SDP method. On the other hand, as *N* increased, the gap between SD/SOCP and WLS becomes smaller, while the gap between the achieved RMSE and the CRLB becomes larger. Figure 4 shows the superior RMSE performance of the proposed methods for all chosen *N*.

Figure 5 shows the simulation results on the effects of the absorption coefficient $\alpha_{f}$ on the RMSE for $\alpha_{f}$ varies from 0.01 dB/m to 0.1 dB/m when $P_{0}$ is known (Figure 5a) and unknown (Figure 5b), where σ=4 dB, and N=10. In this case, u∈[−0.05,−0.5], and v=[0.1151,1.1513]. Similarly, both SD/SOCP methods were superior to the SDP method and WLS method for all $\alpha_{f}$ values. Meanwhile, both SD/SOCP methods varied slightly as $\alpha_{f}$ increased. The SDP method showed a trend of first decline and then rise. The results show that the proposed methods have robust performance for $\alpha_{f}$ varying from 0.01 dB/m to 0.1 dB/m.

To further verify the effectiveness of the proposed methods, it was compared with other discussed methods. We also considered the localization scenario of three-dimensional underwater acoustic wireless sensor networks, where the anchor and target nodes were randomly located within a cube region of 100 m×100 m×100 m. Figure 2b shows an example of a 3D network configuration and estimation accuracy results for the proposed methods. Figure 6 shows the simulation results on the effects of the noise standard derivation σ on the RMSE for σ varying from 1 dB to 5 dB in three-dimensional underwater acoustic wireless sensor networks, where the number of anchor nodes is also fixed at N=10, and the absorption coefficient was fixed at $\alpha_{f}=0.01$ dB/m. By the simulation setups, we had u=−0.05 and v=0.1151 for the approximations in Equations (Equation 11) and (21), respectively. As in Figure 3, in these figures, it can be seen that the proposed methods also showed excellent performance in three-dimensional scenario.

It may be concluded that, although both SD/SOCP-K and SD/SOCP-U are derived under the assumptions of small values for both noise and absorption coefficient, the simulations showed that excellent performance could still be achieved, even for high noise level and large absorption coefficient.

## 6. Conclusions

A mixed SD/SOCP approach has been proposed for the RSS-based target localization problem in UWSNs with and without the knowledge on the target transmit power. In the case of known transmit power, we treat the reference power as a constant and derive a novel non-convex WLS estimator based on appropriate approximations. Then, the WLS estimation problem is transformed into a mixed SD/SOCP for reaching efficiently an optimized solution. In the case of unknown transmit power, we treat the reference power as an additional unknown parameter and propose a combined ML and SD/SOCP iterative algorithm to jointly estimate both the reference power and the target location. In this paper, we only consider a simplified model, and the depth is assumed to be known in advance. Furthermore, Only simulation results are provided to verify the performance of the proposed methods. We hope to extend the present work in a general model, and for the results to be verified by real data in our future work.

## Figures and Tables

**Figure 1 sensors-19-02323-f001:**
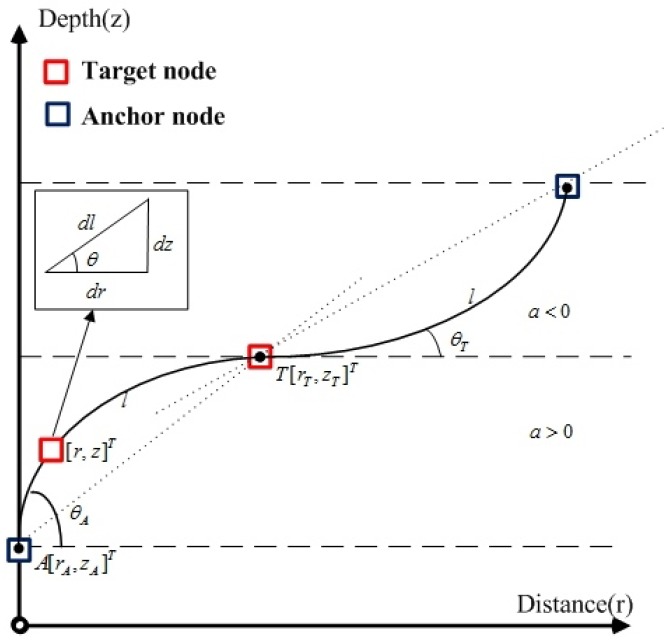
The stratification effect description of a ray between a target node and an anchor node though two adjacent layers.

**Figure 2 sensors-19-02323-f002:**
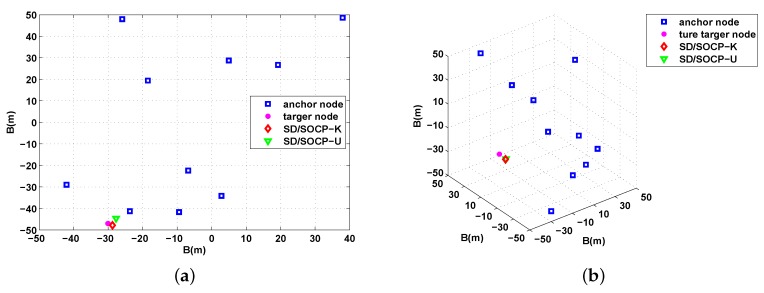
Example of a network configuration and estimation accuracy results for the proposed methods. (**a**) Example of a 2D network configuration and estimation accuracy results for the proposed methods; (**b**) Example of a 3D network configuration and estimation accuracy results for the proposed methods.

**Figure 3 sensors-19-02323-f003:**
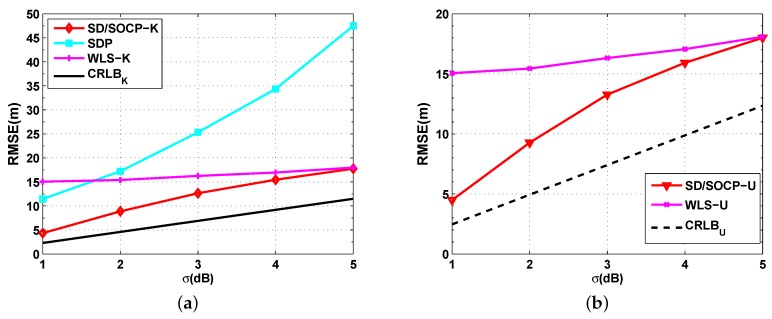
RMSE versus the noise standard deviation σ. (**a**) Simulation results for UWSNs localization when $P_{0}$ is known: RMSE versus the noise standard deviation σ with N=10, and $\alpha_{f}=0.01$ dB/m; (**b**) Simulation results for UWSNs localization when $P_{0}$ is unknown: RMSE versus the noise standard deviation σ with N=10, and $\alpha_{f}=0.01$ dB/m.

**Figure 4 sensors-19-02323-f004:**
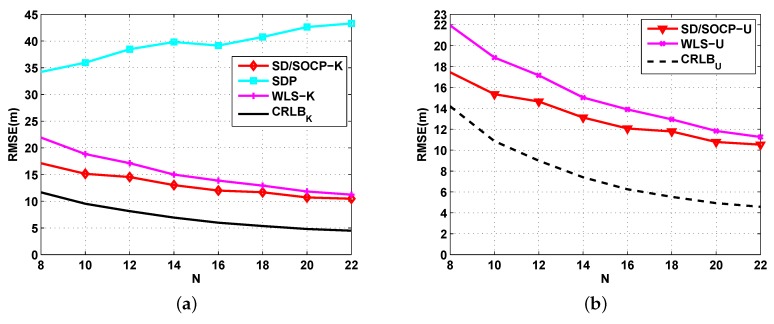
RMSE versus the number of anchor nodes *N*. (**a**) Simulation results for UWSNs localization when $P_{0}$ is known: RMSE versus the number of anchor nodes *N* with σ=4 dB, and $\alpha_{f}=0.01$ dB/m; (**b**) Simulation results for UWSNs localization when $P_{0}$ is unknown: RMSE versus the number of anchor nodes *N* with σ=4 dB, and $\alpha_{f}=0.01$ dB/m.

**Figure 5 sensors-19-02323-f005:**
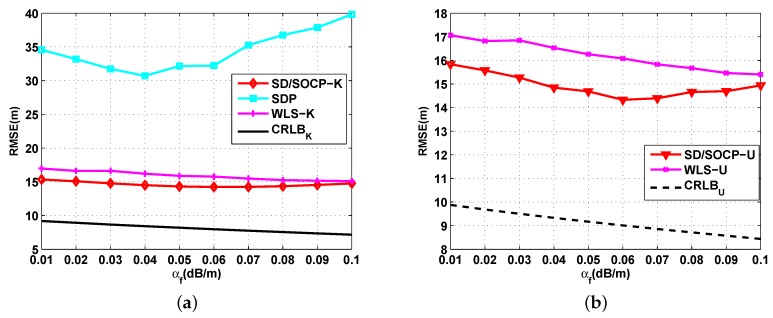
RMSE versus the absorption coefficient $\alpha_{f}$. (**a**) Simulation results for UWSNs localization when $P_{0}$ is known: RMSE versus the absorption coefficient $\alpha_{f}$ with σ=4 dB, and N=10; (**b**) Simulation results for UWSNs localization when $P_{0}$ is unknown: RMSE versus the absorption coefficient $\alpha_{f}$ with σ=4 dB, and N=10.

**Figure 6 sensors-19-02323-f006:**
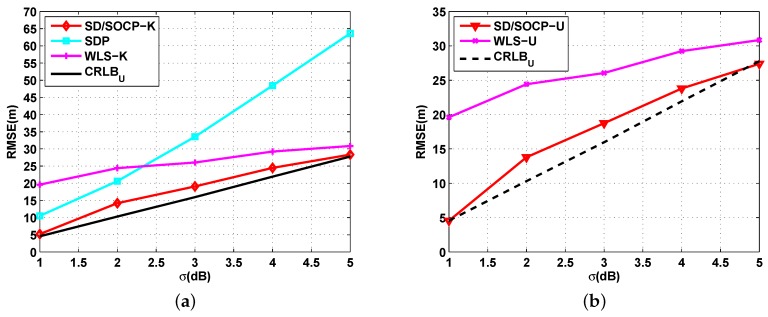
RMSE versus the noise standard deviation σ in 3D space. (**a**) Simulation results for UWSNs localization when $P_{0}$ is known: RMSE versus the noise standard deviation σ with N=10, and $\alpha_{f}=0.01$ dB/m; (**b**) Simulation results for UWSNs localization when $P_{0}$ is unknown: RMSE versus the noise standard deviation σ with N=10, and $\alpha_{f}=0.01$ dB/m.

**Table 1 sensors-19-02323-t001:** The parameters for simulations.

Parameters	$l_{0}$	$P_{0}$	γ	ρ	Mc
**Value**	1 m	−40 dBm	2	0.8	3000

**Table 2 sensors-19-02323-t002:** Summary of the compared methods.

Method	Description
SDP	SDP method in [13]
WLS-K	WLS-K method in [14] for known transmit power
WLS-U	WLS-K method in [14] for unknown transmit power
SD/SOCP-K	Proposed the new SD/SOCP-K method for known transmit power (Equation (19))
SD/SOCP-U	Proposed the new SD/SOCP-K method for unknown transmit power (Equation (28))
CRLB${}_{k}$	Lower limit on the variance of any unbiased estimators for known transmit power
CRLB${}_{U}$	Lower limit on the variance of any unbiased estimators for unknown transmit power

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
