# Peer review of "RSS-Based Target Localization in Underwater Acoustic Sensor Networks via Convex Relaxation"

_sensors, 2019, doi:10.3390/s19102323_

Round 1

Reviewer 1 Report

The authors of this paper present a mixed SD/SOCP approach for the RSS-based target localization problem in UWSNs with and without the knowledge on the target transmit power. Their simulation results have shown a correct performance can still be achieved even for high noise level and large absorption coefficient. This paper has a good mathematical study of the problem, but there are other aspects that should be improved. For example: 1. Section 1 should be divided into 2, the first called introduction where the authors introduce the subject matter of the problem and how it is going to be solved. The second should deal with all related work. In this section the authors must introduce papers related to the topic and show the differences between their work and the previous ones. 2. In this work, the authors show in figure 2 a 2D example. It would be better to show a 3D example, because underwater communications works in 3D. 3. The simulations show the good functioning of the proposals, but when the characteristics of the channel worsen (typical in this type of communications) it can be observed that proposals' results are similar to other previous algorithms. The authors should introduce more simulations to see the goodness of the proposal. They should also add a table comparing their proposal with others. 4. The conclusions should improve and introduce more information of the obtained results.

Author Response

Manuscript ID: sensors-489694

RSS-Based Target Localization in Underwater Acoustic Sensor Networks via Convex Relaxation

We would like to sincerely thank you for arranging the review of our manuscript. Special thanks to the reviewers who have provided very insightful and sensible comments that we feel would considerably improve the quality of our manuscript.

Following your instructions, we have carefully studied and addressed the reviewers’ comments and revised the manuscript accordingly, as you can see from revised submission, together with some misspellings and unclear statements/descriptions corrected and enhanced. Our responses to each reviewer's comments are numbered and listed in this letter, where the reviewers’ original comments are in italic and our responses in plain blue. Please also note that all the cited lines, paragraphs and pages in our response hereunder are referred to the revised manuscript.

We hope that this revision has been improved to a satisfactory and acceptable level and we very much appreciate your consideration for the publication.

Your sincerely,

Shengming Chang, Youming Li, Yu-Cheng He, and Yongqing Wu

Authors' Response to Reviewer 1

Comment R1-1

1.      Section 1 should be divided into 2, the first called introduction where the authors introduce the subject matter of the problem and how it is going to be solved. The second should deal with all related work. In this section the authors must introduce papers related to the topic and show the differences between their work and the previous ones.

Response:

Thanks you for your useful suggestion. We have divided the introduction according to the comments in the revised manuscript.

Comment R1-2

2.      In this work, the authors show in figure 2 a 2D example. It would be better to show a 3D example, because underwater communications works in 3D.

Response:

Thanks for your kind suggestion. We have added a 3D example in Figure 2 (b) in the revised manuscript.

Comment R1-3

3.      The simulations show the good functioning of the proposals, but when the characteristics of the channel worsen (typical in this type of communications) it can be observed that proposals' results are similar to other previous algorithms. The authors should introduce more simulations to see the goodness of the proposal. They should also add a table comparing their proposal with others.

Response:

Thanks for your kind suggestion. We also conduct more simulations in three-dimensional underwater acoustic wireless sensor networks, where the anchor and target nodes are randomly located within a cube region of 100m×100m×100m. Figure 6 shows the simulation results of the discussed methods in three-dimensional underwater acoustic wireless sensor networks. The results also show that the proposed methods show excellent performance in this scenario.   

Comment R1-4

4.      The conclusions should improve and introduce more information of the obtained results.

Response:

Thanks for your kind suggestion. We provide more information on how to improve the proposed results in the future. 

Reviewer 2 Report

This paper is generally well written, good structure, enough contribution. However, there are only some minor issues that need to be addressed:

1- Equations 2, 6 and 7 need to be cited by appropriate references, or it should be explained how they can be obtained. 

2- How the proposed model can work in a real-world scenario. For instance, who is responsible to perform the Algorithm 1 and convex optimisation and how the required information can be collected and disseminated. 

3- In Algorithm 1, the spaces between lines are too short which makes it hard to be read. 

4- It is better to include a table for simulation parameters in the simulation results section. 

5- The simulation results for WLS−K in Figure 3 and 4 are not visible. Maybe it is better to use dashed or dotted lines for them to be more transparent.

Author Response

Manuscript ID: sensors-489694

RSS-Based Target Localization in Underwater Acoustic Sensor Networks via Convex Relaxation

We would like to sincerely thank you for arranging the review of our manuscript. Special thanks to the reviewers who have provided very insightful and sensible comments that we feel would considerably improve the quality of our manuscript.

Following your instructions, we have carefully studied and addressed the reviewers’ comments and revised the manuscript accordingly, as you can see from revised submission, together with some misspellings and unclear statements/descriptions corrected and enhanced. Our responses to each reviewer's comments are numbered and listed in this letter, where the reviewers’ original comments are in italic and our responses in plain blue. Please also note that all the cited lines, paragraphs and pages in our response hereunder are referred to the revised manuscript.

We hope that this revision has been improved to a satisfactory and acceptable level and we very much appreciate your consideration for the publication.

Your sincerely,

Shengming Chang, Youming Li, Yu-Cheng He, and Yongqing Wu

Authors' Response to Reviewer 2

Comment R2-1

1.      Equations 2, 6 and 7 need to be cited by appropriate references, or it should be explained how they can be obtained.  

Response:

Thanks for your insightful comment. More references about the formulas are cited in the revised manuscript.

Comment R2-2

2.      How the proposed model can work in a real-world scenario. For instance, who is responsible to perform the Algorithm 1 and convex optimization and how the required information can be collected and disseminated.

Response:

Thanks for your insightful comment. Due to limited conditions, we are sorry to let you know that we don't have such testbed system at present time. So, we haven't real data for further analysis. We hope to do some small scale experiments in the near future. In fact, our proposed algorithm will be implemented under a centralized processing mode, where, the anchor nodes convey their RSS measurements to the central processor for estimating the target location.

Comment R2-3

3.      In Algorithm 1, the spaces between lines are too short which makes it hard to be read.

Response:

We have adjusted the spaces between lines according to your kind suggestion.

Comment R2-4

4.      It is better to include a table for simulation parameters in the simulation results section.

Response:

Thanks for your kind comment. We provide table I about the simulation parameters in the revised manuscript.

Comment R2-5

5.      The simulation results for WLS-K in Figure 3 and 4 are not visible. Maybe it is better to
use dashed or dotted lines for them to be more transparent
.

 Response:

Thanks for your comment. In the current manuscript, we provide the simulation results in two figures corresponding to known transmission power and unknown transmission power cases. In this way, the simulation results can be visible easily.

Reviewer 3 Report

@page { margin: 0.79in } p { margin-bottom: 0.1in; line-height: 120% }

The paper proposes estimators for the location of an underwater target based on weighed-least-squares formulation, transformed into a mixed semidefinite-second order cone programming (SD/SOCP)  problem.

Two separate cases are considered, depending on whether the source transmit power is known or not. In the latter case, the SD/SOCP problem is iterated with a ML estimator.

My comments follow in order of appearance in the manuscript.

1) Which scenarios are you focusing on for your paper? Specifically what kind of scenario or application does Figure 1 refer to?

2) It would be useful for the reader to plot the sound speed profile used to derive the ray trajectory in Figure 1.

3) The statement before Equation (1) is wrong: the SSP under water is NOT always the constant-gradient one specified in (1). This is at most a local approximation. Please rewrite the sentence before (1) correctly.

4) on page 4, before section 2.2, how many layers do you consider in your layered, constant-sound-speed-per-layer model?

5) your localization algorithm works in 2D. In order to estimate the depth, you assume the use of a depth sensor at the node to be localized. This is only possible in a few cases. Why don't you extend your algorithm to estimate the location of the node in 3D? You already have vector notation throughout the development of your method, so this should not be too difficult.

5) How do you carry out your Monte-Carlo simulations? What varies between across the different realizations?

6) Please proofread the paper carefully: it still contains several typos and awkward expressions.

Author Response

Manuscript ID: sensors-489694

RSS-Based Target Localization in Underwater Acoustic Sensor Networks via Convex Relaxation

We would like to sincerely thank you for arranging the review of our manuscript. Special thanks to the reviewers who have provided very insightful and sensible comments that we feel would considerably improve the quality of our manuscript.

Following your instructions, we have carefully studied and addressed the reviewers’ comments and revised the manuscript accordingly, as you can see from revised submission, together with some misspellings and unclear statements/descriptions corrected and enhanced. Our responses to each reviewer's comments are numbered and listed in this letter, where the reviewers’ original comments are in italic and our responses in plain blue. Please also note that all the cited lines, paragraphs and pages in our response hereunder are referred to the revised manuscript.

We hope that this revision has been improved to a satisfactory and acceptable level and we very much appreciate your consideration for the publication.

Your sincerely,

Shengming Chang, Youming Li, Yu-Cheng He, and Yongqing Wu

Authors' Response to Reviewer 3

Comment R3-1

1.        Which scenarios are you focusing on for your paper? Specifically what kind of scenario or application does Figure 1 refer to?

Response:

Thanks you for your good suggestion. We consider a target localization problem in a 3D underwater acoustic environment in Figure 1, which axis z stands for the depth of the targets, and lateral axis for the distances between target and anchor nodes.

Comment R3-2

2.      It would be useful for the reader to plot the sound speed profile used to derive the ray trajectory in Figure 1.

Response:

Thanks you for your useful suggestion. More information is provided which may help the reader to know the ray trajectory in Figure 1.

Comment R3-3

3.      The statement before Equation (1) is wrong: the SSP under water is NOT always the constant-gradient one specified in (1). This is at most a local approximation. Please rewrite the sentence before (1) correctly.

Response:

We greatly thank the reviewer for the insightful suggestions. From the reference [16],[17],[18], we know that equation (1) is approximately true.

Comment R3-4

4.      On page 4, before section 2.2, how many layers do you consider in your layered, constant-sound-speed-per-layer model?

Response:

We greatly thank the reviewer for the insightful suggestions. Before Section 2.2, we discuss a general model, but in our simulations, only one layer is considered.

Comment R3-5

5.  Your localization algorithm works in 2D. In order to estimate the depth, you assume the use of a depth sensor at the node to be localized. This is only possible in a few cases. Why don't you extend your algorithm to estimate the location of the node in 3D? You already have vector notation throughout the development of your method, so this should not be too difficult.

Response:

We greatly thank the reviewer for the insightful suggestions. Our method is suitable for 2D and 3D cases, only in the simulation part, we assume that the depth of the target node is known, which can be obtained through pressure sensors. Therefore, only plane position x and y coordinates of target location should be estimated. In fact, similar processing is also seen in [31]..

Comment R3-6

6.      How do you carry out your Monte-Carlo simulations? What varies between across the different realizations?

Response:

We greatly thank the reviewer for the insightful suggestions. In our Monte Carlo simulations, 3000 independent runs are obtained to computer the root mean square error (RMSE). Figure 3-5 respectively shows the RMSE versus the noise variance, the number of anchor nodes, and the absorption coefficient.

Comment R3-7

7.      Please proofread the paper carefully: it still contains several typos and awkward expressions.

Response:

Thanks for your kind suggestion. We have polished and enhanced all the grammar, typing, and format errors in the current draft

Round 2

Reviewer 1 Report

The authors have included the changes related to the previous comments.

Author Response

Dear  Reviewer,

We would like to sincerely thank thanks to you and have provided very insightful and sensible comments that we feel would considerably improve the quality of our manuscript. 

Reviewer 3 Report

Re Comment R3-1: 

This is not the kind of answer I was expecting. What you answered can be inferred from the text. What I mean is: in what scenarios is the problem of localizing a target in 3D relevant? For what applications? This is important to understand the feasibility of your performance evaluation in light of the typical deployment that a given application would entail. 

Also note that it is not sufficient to answer the reviewer in the rebuttal letter, you also need to insert new text in the manuscript in order to address the reviewer's comment.

Re Comment R3-2:

Again, not the answer I was expecting. If the ray in Figure 1 is traced under a given sound speed profile (e.g., the constant gradient profile in Eq. (1)), please so state. Otherwise, state what sound speed profile is used.

Re Comment R3-3:

It is not sufficient to cite [16-18] in support of your statement: the approximation in (1) is not a general one, and you need to also summarize the conditions under which a constant-gradient sound speed profile is a good approximation.

Re Comment R3-4:

Having only one layer limits the breadth of your performance evaluation. For fairness, you need to clearly state that you assume a single layer in your simulations.

Re Comment R3-5:

It would be helpful to have a 3D simulation, perhaps in a smaller scenario. This would make it possible for the reader to evaluate if your method really has merit in 3D scenarios as well as in 2D ones.

Re Comment R3-6:

My question was different: what is different between 2 Monte Carlo runs? What is randomly extracted? Also please note that you should insert this detail in the manuscript.

Author Response

Manuscript ID: sensors-489694

RSS-Based Target Localization in Underwater Acoustic Sensor Networks via Convex Relaxation

We would like to sincerely thank you for arranging the review of our manuscript. Special thanks to the reviewers who have provided very insightful and sensible comments that we feel would considerably improve the quality of our manuscript.

Following your instructions, we have carefully studied and addressed the reviewers’ comments and revised the manuscript accordingly, as you can see from revised submission, together with some misspellings and unclear statements/descriptions corrected and enhanced. Our responses to each reviewer's comments are numbered and listed in this letter, where the reviewers’ original comments are in italic and our responses in plain blue. Please also note that all the cited lines, paragraphs and pages in our response hereunder are referred to the revised manuscript.

We hope that this revision has been improved to a satisfactory and acceptable level and we very much appreciate your consideration for the publication.

Your sincerely,

Shengming Chang, Youming Li, Yu-Cheng He, and Yongqing Wu

Authors' Response to Reviewer 3

Re Comment R3-1:

1.  This is not the kind of answer I was expecting. What you answered can be inferred from the text. What I mean is: in what scenarios is the problem of localizing a target in 3D relevant? For what applications? This is important to understand the feasibility of your performance evaluation in light of the typical deployment that a given application would entail.

Also note that it is not sufficient to answer the reviewer in the rebuttal letter, you also need to insert new text in the manuscript in order to address the reviewer's comment.

Response:

Thanks you for your good suggestion. In (1), z denotes the depth, b indicates the sound speed at the water surface, and a is the steepness of SSP depending on the environment of the stratification effect of water medium. Let [xA,yA,zA] and [xT,yT,zT] denote the [rA,zA] and [rT,zT] respectively in 3D space. From Figure 1, we can seen that the UWSNs is 3D, while the ray equations are established in a 2D plane. This because of the cylindrical symmetry around the z axis when the z axis crosses anchor node A. In this case, we can transfer the target localization problem to the 2D plane which includes both nodes and z axis. When the z axis does not cross anchor node A, we still consider it to be a general 3D space. So, Figure 1 shows the general 3D space, We also added this point in detail in the revised manuscript. In order to further illustrate the feasibility of the proposed algorithm in 3D scenarios, Figure 6 also given the RMSE versus the noise standard deviation σ in 3D space in the revised manuscript.

Re Comment R3-2:

2.  Again, not the answer I was expecting. If the ray in Figure 1 is traced under a given sound speed profile (e.g., the constant gradient profile in Eq. (1)), please so state. Otherwise, state what sound speed profile is used.

Response:

Thanks you for your useful suggestion. The ray in Figure 1 is traced by the Snell’s law[18]:

It may help the reader to know the ray trajectory in Figure 1. We also added this point in detail in the revised manuscript.

Re Comment R3-3:

3.      It is not sufficient to cite [16-18] in support of your statement: the approximation in (1) is not a general one, and you need to also summarize the conditions under which a constant-gradient sound speed profile is a good approximation.

Response:

Thanks for your kind suggestion. We have summarized the conditions under which a constant-gradient sound speed profile is a good approximation. We also added this point in detail in the revised manuscript:The main considered in this paper are the stratification effect of water medium. Aiming at the stratification effect of underwater environment, in this paper, we propose a new approach to the RSS-based underwater acoustic localization problem based on the convex relaxation technique in UWSNs. In this approach, the underwater sound speed profile (SSP) is assumed only linearly depth dependent and can be approximated as (1).

Re Comment R3-4:

4.       Having only one layer limits the breadth of your performance evaluation. For fairness, you need to clearly state that you assume a single layer in your simulations.

Response:

Thanks for your kind suggestion. We have explained this point in detail in the revised manuscript :We also assume that only one layer of SSP is considered in the simulations.

Re Comment R3-5:

5.      It would be helpful to have a 3D simulation, perhaps in a smaller scenario. This would make it possible for the reader to evaluate if your method really has merit in 3D scenarios as well as in 2D ones.

Response:

Thanks for your kind suggestion. We have added a 3D simulation of RMSE variation with noise standard deviation in the simulation section. It can be seen in Figure 6: RMSE versus the noise standard deviation σ in 3D space in the revised manuscript.

Re Comment R3-6:

6.      My question was different: what is different between 2 Monte Carlo runs? What is randomly extracted? Also please note that you should insert this detail in the manuscript.

Response:

Thanks for your kind suggestion. Now we understand what your mean. In each Monte Carlo simulation, the anchor and target nodes are randomly located within a square region of 100×100m2. The estimation of random variables (target location) in each Monte Carlo simulation is independent of each other. We have explained in detail in the revised manuscript.

Round 3

Reviewer 3 Report

No further comments.